# Proton Therapy in the Management of Hepatocellular Carcinoma

**DOI:** 10.3390/cancers14122900

**Published:** 2022-06-12

**Authors:** Jana M. Kobeissi, Lara Hilal, Charles B. Simone, Haibo Lin, Christopher H. Crane, Carla Hajj

**Affiliations:** 1Department of Radiation Oncology, School of Medicine, American University of Beirut Medical Center, Beirut 1107, Lebanon; jmk24@mail.aub.edu (J.M.K.); lh54@aub.edu.lb (L.H.); 2New York Proton Center, Department of Radiation Oncology, New York, NY 10035, USA; csimone@nyproton.com (C.B.S.2nd); hlin@nyproton.com (H.L.); 3Memorial Sloan Kettering Cancer Center, Department of Radiation Oncology, New York, NY 10027, USA; cranec1@mskcc.org

**Keywords:** hepatocellular carcinoma, stereotactic body proton therapy, intensity-modulated proton therapy, image guided proton therapy, toxicity

## Abstract

**Simple Summary:**

Radiation therapy is among the locoregional therapy modalities used to treat unresectable or medically inoperable hepatocellular carcinoma (HCC). Proton radiation therapy plays a major role in the treatment of HCC, especially when liver toxicity is a concern. The aim of this review is to provide a concise and comprehensive summary on the use of proton therapy in the management of HCC.

**Abstract:**

Proton radiation therapy plays a central role in the treatment of hepatocellular carcinoma (HCC). Because of the near-zero exit dose and improved sparing of normal liver parenchyma, protons are being used even in challenging scenarios, including larger or multifocal liver tumors, and those associated with vascular tumor thrombus. There is a mounting level of evidence that suggests that protons are superior to photons in terms of survival and toxicity outcomes, specifically the progression to liver failure. A randomized controlled trial comparing protons to photons is currently underway to verify this hypothesis.

## 1. Introduction

Liver cancer is the sixth most common cancer worldwide, making up around 5% of all new cancer cases in 2020 [1]. It is particularly common in East Asia and Africa [1], but its incidence has been increasing in other parts of the world as well. This includes the United States [2], where it now makes up around 2.2% of all new cases, with an estimated 5-year relative survival rate of only 20.3% [1].

Since the majority of cases develop in patients with underlying cirrhosis, the overall performance status is an important factor to consider in addition to the underlying liver function and cancer stage when deciding on management [3]. Early-stage disease can be managed with local curative options, including ablation, resection, or even liver transplantation. Multifocal disease can be treated with chemoembolization or systemic therapy. In advanced stages, systemic therapy is the mainstay of treatment, with palliative care preserved for terminal cases [3]. Radiation falls under the locoregional therapies available for inoperable or unresectable tumors. The NCCN guidelines allow for the use of proton therapy in the management of primary liver tumors [4].

The application of proton therapy in the management of hepatocellular carcinoma (HCC) has been proposed to decrease toxicity, since protons deposit a dose at a specific depth with no exit dose and a minimal scatter to nearby organs [5,6]. With more centers adopting proton therapy, this advanced modality has been the subject of active research, especially over the past few years [7]. In this paper, we present an up-to-date review of the literature on proton radiation treatment for HCC.

## 2. Dosimetric Data

Proton beam therapy (PBT) has been shown to be dosimetrically superior to other photon radiation therapy modalities (XRT) in the management of HCC. Li et al. noted reductions in the mean liver dose (Dmean) and V10–V30 Gy when proton plans were compared to 3D conformal radiation therapy (3D-CRT) for stage I disease (V30: 10.66% [PBT] vs. 21.24% [3D-CRT], *p* < 0.002), and to 3D-CRT and intensity-modulated radiation therapy (IMRT) for stage IIa disease (V30: 22.78% [PBT] vs. 44.01% [3D-CRT] and 37.75% [IMRT], *p* < 0.002) [8]. The proton plans were also superior in sparing other organs at risk (OARs), namely the stomach and the right kidney. Another group of researchers compared PBT to helical (H-)IMRT or volumetric-modulated arc therapy (VMAT) [9]. While all three plans were similar in target coverage, PBT significantly lowered the mean liver dose relative to either photon technique (*p* < 0.05), as well as the V5–V45 and V5–V35 when compared to H-IMRT and VMAT, respectively (*p* < 0.05). In yet another comparison, particle therapy decreased the mean dose received by the normal liver (*p* < 0.05), which translated into a dramatically lower estimated risk of classic radiation-induced liver disease (RILD): 22.3% and 2.3% for IMRT and PBT, respectively (*p* < 0.05) [10].

With an improved sparing of both normal liver and other OARs, protons may allow higher doses to be prescribed to the target volume. This was shown in a planning study that included 30 patients with tumors at risk of requiring radiation dose de-escalation [11]. Intensity-modulated proton therapy (IMPT) and VMAT-rapid arc (VMAT-RA) plans were compared, and a maximum dose of 75 Gy was prescribed in three fractions. Two thirds (20/30) of the VMAT-RA plans violated at least one dose constraint compared to just two plans of IMPT. Thus, the proton radiation plans allowed the maintenance of the prescribed 25 Gy per fraction in most patients.

The dosimetric advantage of protons was also evaluated as a function of tumor size. Toramatsu et al. generated both spot scanning proton (SSPT) and IMRT plans for 10 patients with HCC tumors, ranging in size from 3.4 cm to 16.1 cm [12]. Using the Lyman normal tissue complication probability model, the risk of RILD was estimated to increase dramatically for IMRT plans for tumor diameters beyond 6.3 cm, at which point the risk was 94.5% and 6.2% in the IMRT and SSPT plans, respectively. The authors concluded that protons are especially useful for larger tumors 6.3 cm and above in diameter.

Other researchers further expanded on the interplay between tumor size, tumor location, and liver sparing. They generated different proton- and photon-based SBRT plans with six ”mock” tumors, ranging in size from 1 to 6 cm, in four different locations of the liver: left medial, caudal, dome, and central [13]. The proton plans spared the normal liver to a greater degree and decreased the mean liver dose compared to photons plans (8.4 vs. 12.2 Gy, *p* = 0.01) only for tumors located at the dome or in central locations with diameters of 3 cm and above. In a later study, the researchers included even larger tumors (up to 10 cm) and were able to show that stereotactic body proton therapy (SBPT) plans maintained an adequate target coverage and OAR sparing up to a tumor diameter of 9 cm (vs. 7 cm for photon stereotactic body radiation therapy (SBRT)) [14]. This translates into a lower liver normal tissue complication probability for SBPT plans in cases of larger lesions, particularly if they are located centrally or in the dome of the liver.

## 3. Clinical Data

### 3.1. Early Experience/University of Tsukuba

The earliest experience in applying proton therapy for HCC originated from the University of Tsukuba, Japan. In 1992, Tanaka and colleagues first noted that 11 patients had a reduction in tumor size after proton beam therapy without sustaining serious side effects [15]. Several other reports were subsequently published, further expanding on this application [16,17,18,19,20,21,22,23,24]. As more patients received proton therapy for their liver tumors over the years, more data were collected. By 2009, researchers from the University of Tsukuba were able to summarize their early experiences and develop their own dosing protocols according to tumor location [25]. For tumors within 2 cm of the digestive tract, patients were treated with 77 GyE in 35 fractions. For tumors within 2 cm of the porta hepatis, patients received 72.6 GyE in 22 fractions. Otherwise, they would receive 66 GyE in 10 fractions. With this approach, the 1-, 3-, and 5-year overall survival rates for 318 patients were 89.5%, 64.7%, and 44.6%, respectively [25]. The factors significantly correlated with survival included: liver function, tumor T stage, performance status, and planning target volume. The treatment was relatively well tolerated, with only one patient sustaining grade three gastrointestinal toxicity, manifesting as a colonic hemorrhage. In a later comparison, the researchers found no significant survival difference between the three treatment dosing protocols [26].

### 3.2. Further Clinical Data/East Asia

Proton therapy for HCC remains a research topic of interest to this day in East Asia. In addition to further retrospective data [27,28,29,30], prospective evidence was also generated. An early phase II trial included 30 patients with HCC and liver cirrhosis who received 76 GyE of PBT in 20 fractions [31]. The 2-year local progression free survival (LPFS) and overall survival (OS) rates were 96% and 66%, respectively. Of note, four patients died of proton-induced hepatic insufficiency within 1 year after treatment. Later, Nakayama et al. would show a more favorable toxicity profile with proton therapy [32]. They prospectively followed 47 patients with HCC who received either 72.6 GyE in 22 fractions or 76 GyE in 35 fractions. Only one patient had a grade three bleeding colonic ulcer, even though all patients had their tumors within 2 cm of the GI tract. The resultant 3-year LPFS and OS rates were 88.1% and 50%, respectively. 

The recommended dose and feasibility of hypofractionation using proton therapy have been extensively studied. In a phase I dose escalation study, higher doses were correlated with significantly higher tumor response rates without increasing toxicity [33]. With the different dose levels of 60 GyE/20, 66 GyE/22, and 72 GyE/24, the resulting response rates were 62.5%, 57.1%, and 100%, respectively (*p* = 0.039). The 3-year overall LPFS was 79.9%, and while it tended to increase with the dose, this increase was not statistically significant (*p* = 0.543). In another study, patients with tumors further away from the GI tract (≥2 cm) and who were able to receive a higher dose showed an improved 5-year LPFS and OS (*p* < 0.001) [34]. With these promising results, Kim el al. carried out a phase II prospective clinical trial [35]. The study enrolled 45 patients who had HCC tumors at least 2 cm away from the GI tract (median size: 1.6 cm), in the setting of Child–Pugh A liver function. After receiving 70 GyE in 10 fractions, all the patients responded to therapy and their 3-year LPFS and OS rates were 95.2% and 86.4%, respectively.

The analyses in the studies mentioned above found different combinations of variables to be correlated with survival. The risk factors that were consistently shown to be relevant include the Child–Pugh classification [26,27,34] and tumor stage [28,29,34]. Another factor is the indocyanine-green retention rate at 15 min (ICG R15). Not only was a low ICG R15 associated with improved survival [36], but it also correlated with a lower risk of developing proton-induced hepatic insufficiency [37]. In assessing the optimal time to evaluate outcomes, Kim et al. noted that among their patients who responded to proton therapy, 94% of them did so within 1 year of treatment [38]. The time the patients took to respond, however, did not affect recurrence or survival, and the researchers suggested that should salvage treatments be desired, they can be postponed by up to 18–24 months.

### 3.3. Clinical Data/United States

The first report on proton therapy for HCC in the Western world originated from Loma Linda University [39]. After publishing preliminary results in 2004, Bush et al. continued to enroll patients in their phase II clinical trial and ultimately reported the outcomes of 76 patients diagnosed with HCC and liver cirrhosis [40]. Thirty-five patients fulfilled the Milan criteria, and of those, eighteen underwent liver transplantation. The latter had a significantly greater 3-year survival rate compared to those who were not transplanted (70% vs. 10%, *p* < 0.001). Upon multivariate analysis, the only significant factor impacting the survival of the overall population was the Milan criteria (*p* = 0.001). This was followed by a multi-institutional phase II trial, which included 44 patients with HCC followed up for a median duration of 19.5 months [41]. Patients with central tumors (i.e., within 2 cm of the porta hepatis) received 58.05 GyE in 15 fractions, whereas the remaining patients received 67.5 GyE for their peripheral tumors. The resultant 2-year PFS and OS rates were 39.9% and 63.2%, respectively. In this trial, no grade 3+ toxicities were noted apart from a single case of grade 3 thrombocytopenia. Later, retrospective analyses [42,43] showed similar 2-year OS rates (54–62%), as well as a significant correlation between a higher biologic effective dose (BED) and improved survival [43]. Similarly, patients enrolled on the Proton Collaborative Group registry receiving a BED > 75.2 GyE had a significantly higher 1-year local control (LC) rate than those with a lower BED (95.7% vs. 84.6%, *p* = 0.029) [44]. More recently, IMPT in 37 patients with HCC yielded promising results, as it proved to be both safe (only one case of grade three toxicity) and effective (1-year LC and OS rates of 94% and 78%, respectively) [45]. Of note, replanning was required in eight patients to improve target coverage. 

### 3.4. Reirradiation

Upon retrospective analysis of 27 HCC patients with at least two courses of PBT, Hashimoto and colleagues noted a 5-year OS rate of 55.6% [46]. With median doses of 62 Gy and 66 Gy for the first and second courses, respectively, five patients developed grade 3+ toxicity, two of them being acute hepatic failure. Based on the toxicity profile, the researchers observed that reirradiation with PBT is safer for patients with peripheral tumors and a score of Child–Pugh A. Oshiro et al. took it one step further, including patients with up to four courses of radiation to the liver [47]. The 83 included patients received a median total dose of 70.5 GyE and had a 5-year OS rate of 49.4%, with no reported cases of RILD. 

### 3.5. HCC with Vascular Tumor Thrombosis

The combination of HCC with portal vein tumor thrombosis (PVTT) is a known poor prognostic factor [48], and several reports have looked to proton therapy as a possible treatment modality. The first to do so were Hata and colleagues, who showed a 5-year PFS rate of 24% in 12 patients receiving 50–72 Gy of PBT to both their liver tumors and PVTT [49]. Sugahara et al. reported a similar 5-year LPFS rate of 20% and noted improved survival for those whose thrombosis responded to radiation [50]. The prognostic value of the PVTT response was further shown by other groups of researchers [51,52]. For example, in one study, responders to radiation proved to have significantly higher 1-year LPFS and OS rates at 85.6% and 80%, respectively, versus those who did not respond (51.3% and 25%, *p* < 0.05) [52]. HCC can also be associated with thrombosis of the inferior vena cava. Mizumoto et al. first reported the cases of three patients with IVC thrombosis who survived for more than 1 year after receiving PBT [53]. In a more recent analysis, the 1-, 2-, and 3-year OS rates of 21 HCC patients with IVC thrombosis were 62%, 33%, and 19%, respectively [54]. The feasibility of proton therapy in the case of vascular thrombosis was also demonstrated in a prospective study [55] and while using a risk-adapted PBT regimen with simultaneous integrated boost [56]. In a recent systematic review and meta-analysis, particle therapy, including protons, led to a significantly higher 1-year OS rate when compared to CRT or SBRT [57]. 

### 3.6. Large Tumors

Larger tumors pose a greater challenge in management. In a retrospective analysis of 22 patients with large liver tumors (median size: 11 cm), proton therapy was effective, yielding a tumor control rate of 87% at 2 years, with no late grade 3+ toxicity [58]. The 2-year OS rate in this analysis was 36%, almost halfway between the figures reported in other studies: 14% [59] and 52.4% [60]. Of note, the patients in the latter study were all classified as Child–Pugh A and had relatively smaller tumors (median size: 9 cm).

### 3.7. Poor Liver Function

Baseline liver function is a significant factor impacting both the available therapeutic options and survival. Hata et al. offered proton therapy to 19 HCC patients with Child–Pugh class C cirrhosis [61]. No grade 3+ toxicities were noted. The resultant 2-year OS rate was 42%, with prognostic factors being patients’ performance status and Child–Pugh score. The same group of researchers later offered a single fraction of 24 Gy to three patients with ascites [62]. Tumor location was confirmed with a CT scan just before each fraction. The results were favorable in two of the patients, as they were disease-free at 13 and 30 months, respectively. The third patient, however, sustained a rupture of the esophageal varices 6 months post radiation and subsequently died.

Table 1 summarizes the key studies on the use of proton beam therapy for HCC in the settings of re-irradiation, vascular thrombosis, large tumor size, or poor liver function.

### 3.8. Toxicity from PBT

A retrospective analysis of 300 patients with HCC noted only minor, transient variations in liver and biliary enzymes over the course of PBT [63]. Another retrospective analysis observed radiologic biliary abnormalities in only 7.2% of patients undergoing PBT with doses of 75 Gy RBE and above. The abnormalities were not attributed to proton therapy, however, since they either occurred outside the field of radiation or in the setting of intrahepatic disease progression [64]. In a larger analysis of 136 patients status post-PBT, the rate of RILD was 14%, and it was found to be correlated with the ratio of unirradiated liver volume to the standard liver volume (ULV/SLV), gross tumor volume, and the Child–Pugh classification [65]. Given the predictive value of the ULV/SLV, the authors postulated lower limits of 50% and 60% for Child–Pugh A and Child–Pugh B patients, respectively, below which radiation is absolutely contraindicated due to the risk of RILD. Other studies also showed the relevance of the irradiated liver ratio [66,67,68]. 

Besides liver toxicity, chest wall toxicity is also a concern. One study noted that out of all the irradiated, contoured ribs, 8.7% sustained fractures [69]. The volume of ribs receiving more than 60 Gy RBE, V60 was a significant predictive parameter, with a cut-off of 4.48 cm^3^. Another study reported a rate of grade three chest wall pain of 11%, especially when the V47, V50, and V58 were above 20, 17, and 8 cm^3^, respectively [70]. Both studies included patients with HCC undergoing a hypofractionated PBT regimen.

### 3.9. Respiratory-Gated/IGPT

Utilizing 4D-CT to administer respiratory-gated PBT was shown to be feasible by several groups. Hong et al. initially reported the outcomes of 15 patients with liver tumors, noting a 2-year OS rate of 40% [71]. While the latter study included both fiducial marker insertion and respiratory-gated treatment, others showed the feasibility of therapy even without fiducials [72] or with abdominal compression [73]. Mizuhata et al. considered 40 patients known to have HCC within 2 cm of the gastrointestinal tract [72]. After undergoing PBT at a dose of 60–80 CGE, only two patients had grade 3+ toxicity, and the 2-year OS rate was 76%. Similarly, Shibata et al. reported on 29 patients with large HCC tumors (>5 cm) who underwent respiratory-gated PBT at 76 CGE in 20 fractions [72]. Only four grade 3 adverse events were reported, and the 2-year OS rate was 61%. 

In an attempt to increase treatment precision, Itawa et al. applied image-guided proton therapy (IGPT) in the management of 45 patients with operable or ablation-eligible tumors, using CT and MRI scans [74]. With a dose range of 66–72.6 Gy (RBE), the patients tolerated the radiation course relatively well, and the resultant 2-year OS rate was 84%. In another study, IGPT was feasible in an elderly population aged 80 years and above [75]. There was no grade 2 or higher RILD, and the patients’ quality of life was preserved throughout treatment, with a 2-year OS rate of 76%. 

### 3.10. Comparison: Different Treatment Modalities

Yoo et al. compared the clinical outcomes after either the passive scattering (PS) or pencil beam scanning (PBS) proton techniques [76]. In this retrospective analysis, 103 patients with HCC were propensity score matched (33 PBS vs. 70 PS), but no differences in outcomes were noted. The population had a 2-year OS rate of 86.4% with similar rates of RILD development (PBS: 3%, PS: 2.9%, *p* = 1). 

Proton therapy was compared to other local treatment modalities, such as trans arterial chemoembolization (TACE), radiofrequency ablation (RFA), and surgical resection. Bush et al. carried out a randomized clinical trial, in which 69 HCC patients received either PBT (n = 33, dose: 70.2 Gy/15) or TACE (n = 36) [77]. While PBT resulted in superior 2-year LC and PFS rates, statistical significance was not reached (LC: 88% vs. 45%, PFS: 48% vs. 31%, *p* = 0.06), and no difference in the 2-year OS rate was noted (59%). PBT did, however, significantly reduce the duration of hospitalization compared to TACE (total number: 24 days vs. 166 days, *p* < 0.001). 

In another phase III non-inferiority randomized trial, 144 patients with no more than two small, recurrent HCC tumors were recruited and randomized to receiving either PBT or RFA, while allowing crossover in the case of infeasibility [78]. PBT proved to be non-inferior, as the 2-, 3-, and 4-year LPFS rates satisfied the predetermined noninferiority criteria. 

Moving on to operable tumors, Tamura et al. compared PBT and surgical resection [79]. Patients with single nodular HCC tumors of maximal diameter 10 cm and no vascular invasion were selected (n = 31 PBT vs. n = 314 surgical resection). Those who underwent surgery had a significantly longer median survival duration (104.1 vs. 64.6 months, *p* = 0.008) in addition to a longer, though insignificant, relapse free survival time (33.8 vs. 14.0 months, *p* = 0.099).

### 3.11. Comparison: PBT vs. XRT

There are no published randomized controlled data comparing proton to photon radiation treatment for HCC. The earliest comparison was through a systematic review and meta-analysis in 2015 [80]. After reviewing the outcomes of 73 cohorts, the authors noted improved oncologic outcomes for patients treated with charged particle therapy when compared to conventional radiotherapy (pooled OS, PFS, and LC). Such outcomes were, however, comparable to those of SBRT. Charged particle therapy also had lower toxicity rates. Further comparisons were carried out later in a retrospective manner, suggesting the superiority of PBT. Sanford et al. reported on the outcomes of 133 patients with unresectable HCC who received either PBT or photon therapy [81]. The PBT group had a longer 2-year OS rate (59.1% vs. 28.6%, adjusted HR = 0.47, *p* = 0.008) and notably lower odds of developing non-classic RILD (OR: 0.26, *p* = 0.03). Notably, developing the latter was itself associated with a worse OS (adjusted HR: 3.83, *p* < 0.001), suggesting that the improved survival with protons may be linked to better liver sparing with proton RT, and less risk of progression to liver failure. This was confirmed by a propensity-matched analysis carried out by Cheng et al. [82]. Besides the protons’ survival benefit (HR: 0.56, *p* = 0.032) and lower risk of RILD (11.8% vs. 36%, *p* = 0.004), PBT delivered a higher BED than the photon treatment (median: 96.56 Gy RBE vs. 62.5 Gy, *p* < 0.001). With that being said, a retrospective analysis compared PBT (n = 71, dose: 98 Gy) and photon SBRT (n = 918, dose: 100 Gy) in patients with HCC [83]. The proton group had longer 1-year (76.5% vs. 64.3%) and 3-year (36.7% vs. 30%) overall survival rates (*p* = 0.01). Of note, the difference in survival in such studies can be due to the lower rates of toxicity with proton therapy and to the heterogeneity of patients’ characteristics. The patients were not similar in terms of baseline characteristics, and those who received protons were more likely to be white and have larger tumors among other predictive factors. Randomized trials are needed to better evaluate survival outcomes. 

## 4. Prospective Trials

Given the promising retrospective data, a phase III randomized trial by NRG Oncology (NRG-GI003) is currently recruiting patients with unresectable or locally recurrent HCC to receive either proton or photon therapy. The primary outcome is overall survival [84]. Another phase III randomized trial will be comparing local control rates after PBT or RFA [85] (Table 2). The remaining studies currently underway are phase II single group assignment studies evaluating: 2-year OS in unresectable HCC [86], 1-year OS in HCC with PVTT [87], and the rate of RILD 4 months after PBT [88]. One phase II trial will be specifically looking into the toxicity profile of SBPT administered in five fractions to stage I-IIIB or recurrent HCC [89]. 

## 5. Evolving Systemic Therapy Options for HCC

While sorafenib monotherapy was the mainstay systemic treatment for patients with advanced unresectable HCC for more than a decade, novel agents have recently been reported to produce a paradigm shift in management [90]. Immune-based combinations have reported superior results in advanced HCC, as witnessed by the IMbrave150 trial that showed improved survival with the use of atezolizumab combined with bevacizumab as compared to sorafenib [91]. The landscape of new immune-based combinations for advanced HCC continues to expand. Abou-Alfa et al., in a large, randomized phase III HIMALAYA trial, reported that durvalumab was non-inferior to sorafenib with a favorable safety profile. The combination of durvalumab plus tremelimumab showed a superior efficacy and favorable benefit-risk profile compared to sorafenib as a first-line treatment for unresectable HCC [92]. Based on the promising results of early-phase clinical trials, pembrolizumab plus lenvatinib also has potential as a novel treatment option in this setting [93]. As suggested by Rizzo et al., investigating the predictors of response to immune checkpoint inhibitors, such as programmed death ligand 1 (PD-L1), gut microbiome, microsatellite instability (MSI), and tumor mutational burden (TMB) is important to allow the proper selection of HCC patients that could derive the most benefit from immunotherapy [90].

For patients with advanced HCC in the setting of Child–Pugh B, De Lorenzo et al. retrospectively analyzed data in a multicenter study that showed that treatment with metronomic capecitabine is a safe option for patients with Child–Pugh B-HCC. Its potential antitumor activity warrants prospective evaluations [94].

There are currently several ongoing trials (e.g., NCT03482102 and NCT03203304) on the combination of immunotherapy with radiation therapy for HCC, mainly using photon-based RT. In the setting of the changing treatment paradigm for unresectable HCC, future studies on radiation therapy should take into consideration the evolving systemic therapy options and their impact on outcomes. 

## 6. Discussion and Conclusions

Radiation therapy is an important locoregional treatment modality used in patients with unresectable HCC. Local control of HCC in the liver is crucial to slow or prevent disease progression, and subsequent progression to liver failure. Although modern series of photon stereotactic ablative RT have shown excellent local control outcomes in the treatment of HCC [95], protons have dosimetric advantages, especially in certain clinical scenarios. There is a growing body of evidence supporting the use of proton therapy for HCC [26,46,47,55,57,58,67]. Multiple dosimetric studies have shown its advantage in sparing the nearby organs at risk, namely the normal liver parenchyma, and subsequently decreasing the risk of radiation-induced liver disease [10]. This is particularly important in the setting of treating large tumors (more than 6–7 cm in size) and especially for tumors located centrally or in the dome of the liver. Proton therapy can also be advantageous over photon-based therapy in patients with relatively poor liver function (Child–Pugh B) or a small normal liver reserve, such as patients heavily pretreated by other local modalities. Another scenario where proton therapy could offer an advantage over photon-based therapy is in the sequential or synchronous treatment of multifocal HCC where we are usually limited by the dose received by the remaining normal liver volume. 

At our institution in New York, we have been treating patients with HCC with either intensity-modulated radiation therapy (IMRT) or proton radiation therapy. Patients are allocated to either modality on a case-by-case basis. Protons are strongly considered for patients with large HCC tumors, multifocal disease, or a Child–Pugh score of ≥B7. Re-irradiation and high tumor/liver volume ratio, where meeting volumetric and mean liver dose constraints cannot be achieved with IMRT, are also indications for proton RT. Our patients are treated with ablative doses of radiation therapy (BED > 100 Gy) using either 5, 10, 15, or 25 fractions. Eligible patients for ablative radiation therapy are those with tumors >1 cm from stomach/small bowel who meet the liver constraints. Lower doses may be considered for HCC to meet liver constraints.

Although limited by the heterogeneity of patient characteristics in nonrandomized trials, proton therapy has been associated with a lower toxicity and improved survival compared to photon-based modalities during retrospective analyses. Further studies are currently underway, including a randomized trial of protons versus photons for HCC, to better evaluate outcomes and inform clinical decisions.

## Figures and Tables

**Table 1 cancers-14-02900-t001:** Key studies on the use of proton beam therapy for HCC in the settings of re-irradiation, vascular thrombosis, large tumor size, or poor liver function.

Authors	Year	N	CTP Score		Intervention	OS	PFS	Toxicity
**Re-Irradiation**								
Hashimoto et al. [46]	2006	27	A	21	2+ courses of PBT 1st course: 62 Gy in 16 fr 2nd course: 66 Gy in 16 fr	**5-year**		**Grade 3+**
			B	6		55.6%	NA	5/27
Oshiro et al. [47]	2017	83	A	73 *	Up to 4 courses of PBT Median total dose: 70.5 GyE	**5-year**		**Acute Grade 3+ or RILD**
			B	10		49.4%	NA	None
**Vascular Thrombosis**								
Hata et al. [49]	2005	12	A	9	HCC with PVTT Median dose: 55 Gy in 10–22 fr	**NA**	**5-year**	**Grade 3+**
			B	3		NA	24.0%	None
Sugahara et al. [50]	2009	35	A	28	HCC with PVTT Median dose: 72.6 GyE in 22 fr	**5-year**	**5-year LPFS**	**Grade 3+**
			B	7		21.0%	20.0%	3/35 (hematologic)
Lee et al. [52]	2014	27	A	18	HCC with PVTT Median dose: 55 GyE in 20–22 fr	**1-year**	**1-year LPFS**	**Grade 3+**
					**PVTT responders versus** **PVTT non-responders**	80.0%	85.6%	None
			B	9		25.0%	51.3%	
					***p*-value**	<0.05	<0.05	
Kim et al. [56]	2017	41	A	36	PBT + SIB to HCC and TVT 50–66 GyE in 10 fr + 30 GyE in 10 fr	**2-year**	**2-year LPFS**	**Grade 3+**
			B	3		51.1%	88.1%	None
Hashimoto et al. [55]	2017	34	A	20	HCC with PVTT or IVCTT Median BED: 81.3 GyE	**1-year**		**Grade 3+**
			B	12		55%	NA	1 G3 acute dermatitis 1 G3 liver enzyme elevation
			C	2				
Sekino et al. [54]	2019	21	A	12	HCC with IVCTT Median dose: 72.6 Gy (RBE) in 22 fr	**3-year**		**Grade 3+**
			B	9		19.0%	NA	None
**Large Tumors**								
Sugahara et al. [58]	2010	22	A	11	Median tumor size: 11 cm Median dose: 72.6 GyE in 22 fr	**2-year**	**2-year**	**Late Grade 3+**
			B	11		36.0%	24.0%	None
Kimura et al. [60]	2017	24	A	24	Median tumor size: 9 cm Median dose: 72.6 GyE in 22 fr	**2-year**		**Grade 3+**
			B	0		52.4%	NA	2 cases of dermatitis
Nakamura et al. [59]	2018	9	A	6	Median tumor size: 15 cm Median dose: 72.6 GyE in 22 fr	**2-year**		**Grade 3**
			B	3		14.0%	NA	2 / 9
**Poor Liver Function**								
Hata et al. [61]	2006	19	C	19	Median dose: 72 Gy in 16 fr	**2-year**	**2-year**	**Grade 3+**
						42.0%	42.0%	None

*—Distribution of CTP scores applied to patients before the first and second radiation courses. CTP: Child–Turcotte–Pugh score; RILD: radiation-induced liver disease; PVTT: portal vein tumor thrombosis; TVT: tumor vascular thrombosis; IVCTT: inferior vena cava tumor thrombosis; LPFS: local progression free survival; PBT + SIB: proton beam therapy with simultaneous integrated boost; fr: fraction.

**Table 2 cancers-14-02900-t002:** Main ongoing and published studies comparing outcomes and toxicities associated with the use of proton radiation therapy versus other local modalities for the treatment of HCC.

Authors	Year	Study Design	N	Comparison	OS	PFS	Toxicity
Hepatocellular Cancer							
Bush et al. [77]	2016	Randomized trial	69		**2-year**	**2-year**	**Hospitaliza-tion**
				**PBT**	59%	48%	24 days
				**TACE**		31%	166 days
				*p*-value	N.S.	0.06	<0.001
Kim et al. [78]	2020	Randomized phase III non-inferiority trial	144		**2-year**	**2-year**	**Grade 3**
				**PBT**	91.7%	31.9%	0%
				**RFA**	90.3%	31.9%	16.1%
				*p*-value	0.821	0.958	<0.001
NCT02640924 [85]	2021 (est.)	Randomized phase III trial	166 (est.)		**Up to 3–5 years In progress**		
				**Proton RT**			
				**RFA**			
Tamura et al. [79]	2020	Retrospective	345		**5-year**	**5-year RFS**	**Grade 3+**
				**PBT**	51.1%	31.1%	3.2%
				**Surgery**	75.8%	37.3%	10.8%
				*p*-value	0.008	0.099	0.343
Yoo et al. [76]	2020	Retrospective	103		**2-year**	**2-year EHPFS**	**RILD**
				**PS-PBT**	87.1%	74.0%	2.9%
				**PBS-PBT**	83.2%	55.5%	3.0%
				*p*-value	0.766	0.345	1
Sanford et al. [81]	2019	Retrospective	133		**2-year**		**Nonclassic RILD**
				**Proton**	59.1%	NA	OR: 0.26
				**Photon**	28.6%	NA	
				*p*-value	NA		0.03
Hasan et al. [83]	2019	Retrospective	989		**1-year**		
				**PBT**	76.5%	NA	NA
				**SBRT**	64.3%	NA	NA
				*p*-value	0.01		
Cheng et al. [82]	2020	Retrospective	110				**RILD**
				**Proton**	HR: 0.56	NA	11.8%
				**Photon**		NA	36.0%
				*p*-value	0.032		0.004
**NRG-GI003** [84] NCT03186898	2029 (est.)	Randomized phase III trial	186 (est.)		**Up to 5 years In progress**		
				**Proton**			
				**Photon**			

TACE: transarterial chemoembolization, RFA: radiofrequency ablation, RFS: relapse free survival, EHPFS: extra-hepatic progression-free survival, RILD: radiation induced liver disease, PS: passive scattering, PBS: pencil beam scanning, OR: odds ratio, HR: hazard ratio.

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
