# Peer review of "Proton Therapy in the Management of Hepatocellular Carcinoma"

_cancers, 2022, doi:10.3390/cancers14122900_

Round 1
Reviewer 1 Report
An interesting review discussing an emerging topic in HCC management. However, some changes are necessary in my opinion:
- Figure 1 is not clear and should be modified, adding the main clinical data of each trial
- The changing scenario of HCC management should be better discussed, and some recent papers assessing this topic should be included (PMID: 34429006; PMID: 29968763 ; PMID: 34167433 )
- A linguistic revision is necessary, by using a professional service.
- A more personal perspective should be included.
We recommend major changes.
Reviewer 2 Report
The authors present a thorough review on the employment of proton beam therapy to treat hepatocellular carcinoma. The authors make a convincing case about the advantages of proton beam therapy over photon radiotherapy in terms of the localized radiation dose of the first type of treatment. The authors pay particular attention onto the advantages of proton beam therapy for hepatocellular carcinoma when alternative treatments (ablation, resection, liver transplantation) are impractical. The authors have consulted an extensive body of scientific and clinical literature
to write their review. Table 1 and Figure 1 provide a suitable synopsis of all the reviewed materials.
The manuscript is well written in terms of English language, presentation and concision. I think that this manuscript is a valuable contribution to discern the relevance of proton beam therapy to treat hepatocellular carcinoma. I am glad to recommend the publication of this manuscript in Cancers in its present for form.
Author Response
Thank you for the review and the comments.
Reviewer 3 Report
Authors show current data supporting proton therapy for HCC.
In my view, OS must not be considered an endopoint of interest in comparing proton beam therapy with different local treatments. Given the absence of an high quality direct comparison (randomized trial), such outcome is impacted by heterogeneity in patients characteristics more than therapy impact.
The manuscript lacks completely of a discussion section. Discussion and interpretation of data available is crucial in such a scenario. Hypothetical proton therapy advantages must be better highlighted, concerning clinical impact of better healthy liver sparing (es. sinchronous or sequential treatment of multi-focal HCC).
Authors in my view in the discussion section may also underline critical issue in data interpretation between photons and protons. In particular, taking into account the excellent local control obtained in modern series of HCC treated with Photon SBRT), who are candidates in which proton therapy could be of benefit? (high volume HCC, multi-focal HCC or poor liver function?)
In conclusion, the paper is heavily immature to show a contemporary picture of proton therapy for HCC
Round 2
Reviewer 1 Report
The authors addressed all the queries and issues we raised.
We recommend Acceptance.
Reviewer 3 Report
Authors meet my goals in modifying paper.